# Perceived Social Support and Health Care Spending as Moderators in the Association of Traditional Bullying Perpetration with Traditional Bullying and Cyberbullying Victimisation among Adolescents in 27 European Countries: A Multilevel Cross-National Study

**DOI:** 10.3390/ijerph21070863

**Published:** 2024-06-30

**Authors:** Patrick Chanda, Masauso Chirwa, Ackson Tyson Mwale, Kalunga Cindy Nakazwe, Ireen Manase Kabembo, Bruce Nkole

**Affiliations:** 1Department of Social Work and Sociology, School of Humanities and Social Sciences, University of Zambia, Great East Road Campus, Lusaka P.O. Box 32379, Zambia; patrick.chanda@unza.zm (P.C.); ackson.mwale@unza.zm (A.T.M.); ireen.kabembo@unza.zm (I.M.K.); 2School of Graduate Studies, Lingnan University, 8 Castle Peak Road, Tuen Mun, Hong Kong, China; 3Department of Social Policy and Social Security Studies, Hochschule Bonn-Rhein-Sieg University of Applied Sciences, Grantham Allee 20, 53757 Sankt Augustin, Germany; 4Department of Psychology, School of Humanities and Social Sciences, University of Zambia, Great East Road Campus, Lusaka P.O. Box 32379, Zambia; kalunga.nakazwe@unza.zm; 5Department of Sociology and Social Policy, Lingnan University, 8 Castle Peak Road, Tuen Mun, Hong Kong, China; 6Ministry of Education, Kasama P.O. Box 410175, Zambia; brucechileshe1@gmail.com

**Keywords:** adolescents, cyberbullying, health care spending, social support, traditional bullying

## Abstract

Existing research has documented the association between bullying perpetration and bullying victimisation. However, it is still unclear how different sources of social support moderate the association between bullying perpetration and bullying victimisation at a cross-national level. Using multilevel binary logistic regression models, this study examined the moderating role of public health care spending and perceived social support (i.e., family and teacher support) in the association between traditional bullying perpetration and victimisation by traditional bullying and cyberbullying among adolescents across 27 European countries. Country-level data were combined with 2017/18 Health Behaviour in School-aged Children (HBSC) survey data from 162,792 adolescents (11-, 13-, and 15-year-olds) in 27 European countries. Results showed that adolescents who perpetrated traditional bullying had a higher likelihood of being victimised by traditional bullying and cyberbullying than adolescents who did not bully others. Results also indicated that the magnitude of the positive association between traditional bullying perpetration and victimisation by traditional bullying and cyberbullying was mitigated among adolescents with more family, teacher, and public health care support. These findings support the notion that multilayered systems of social support could play a vital role in bullying prevention and intervention strategies to address bullying among adolescents.

## 1. Introduction

The literature indicates that bullying victimisation is a pervasive public health and social problem affecting school-going children and adolescents globally [1,2,3]. Research reveals that victims of both traditional bullying and cyberbullying are more likely to experience psychological problems such as sadness, feelings of hopelessness, low self-esteem, high levels of anxiety, depression, and suicidal ideation [4,5]. Moreover, victims of bullying are more likely to report severe psychological problems, such as post-traumatic stress disorder (PTSD) and adjustment disorders, compared to non-victims [6]. Moreover, victims of bullying are more likely to report health risk behaviours, such as smoking cigarettes, drug use, alcohol consumption, and drunkenness [4,7], and aggressive behaviours, including physical fighting and bullying perpetration [4,7]. Evidence indicates that 45.8% of adolescents experienced both traditional bullying and cyberbullying victimisation in European and North American countries over time, of which 46.5% were male adolescents and 45.3% were females [8].

Although scholars have defined bullying victimisation differently, four key elements that reoccur are repeated aggressive behaviours over time, harmful behaviours, the intention of causing harm to the victim, and power imbalance [9]. While traditional (face-to-face) bullying victimisation among school-going adolescents takes place offline in social environments such as schools and communities [10], cyberbullying victimisation occurs online due to the rapid growth of technology, resulting in the intense and frequent use of the internet, social media, and other digital platforms by adolescents across countries [11]. The term “traditional bullying victimisation” refers to repeated aggression and intentional harm or disturbance experienced by the less powerful person or group through negative verbal or psychological, physical, and relational actions perpetrated by a more powerful person or group [10,12]. In contrast, cyberbullying victimisation is defined as exposure to diverse types of repeated aggression and intentional harm (i.e., psychological and relational harm) and humiliation perpetrated by the more powerful person or group through electronic media, including cyber stalking, name-calling, rumours, or gossip sent via e-mails, hurtful text messages, unpleasant photos or videos posted/disseminated on social media platforms, and exclusion from online communities or social networks [11].

Bronfenbrenner’s [13] social–ecological model holds that adolescents’ aggressive behaviours (such as traditional bullying perpetration) place adolescents at high risk of peer victimisation within social environments, such as the school, community, and societal contexts [14,15,16]. In line with the social–ecological model, existing studies suggest that traditional bullying perpetration can influence bullying victimisation and vice versa, or that the association might be reciprocated or bidirectional [17,18,19,20]. On the one hand, studies have shown that bullying victimisation substantially influences bullying perpetration [20,21]. For example, Shetgiri and colleagues [21] reported that bullying victimisation was a risk factor for bullying perpetration. Likewise, a study by Walters [20] found that bullying victimisation predicted bullying perpetration. Similarly, a study by Walters and Espelage [19] reported that bullying victimisation influenced bullying perpetration. On the other hand, the literature indicates that adolescents who frequently engage in bullying perpetration are at increased risk of bullying victimisation [17,18,20]. For instance, Chu and associates [18] found that traditional bullying perpetration influenced cyberbullying victimisation. Chan and Wong [17] found that bullying perpetration was robustly and positively correlated with traditional bullying victimisation. A study by Cho [22] found that adolescents’ involvement in bullying perpetration was associated with higher risks of bullying victimisation. Likewise, Park and Cho [23] reported that bullying others was significantly associated with traditional bullying victimisation among adolescents. Using the Health Behaviour in School-aged Children (HBSC) data, Deryol and colleagues [24] found that bullying perpetration was associated with bullying victimisation. Relatedly, a study by Piquero et al. [25] found that adolescents who perpetrate bullying are more likely to be victims of bullying. However, existing literature suggests that the association between traditional bullying perpetration and victimisation by cyberbullying and traditional bullying remains unclear at the cross-national level. Despite that the association between bullying perpetration and bullying victimisation has been supported by empirical evidence from single-country studies [15,17,26] and theoretical perspectives, such as the social–ecological model [15,26], few multilevel cross-national studies have examined the linkage between bullying perpetration and bullying victimisation [24]. Thus, it is still not evidently known how traditional bullying perpetration predicts victimisation by traditional bullying and cyberbullying among adolescents across 27 European countries. Moreover, until now, multilevel cross-national studies that used the HBSC data to examine the association between traditional bullying perpetration and victimisation by cyberbullying and traditional bullying have been lacking [24]. To that effect, evidence is needed to confirm how traditional bullying perpetration relates to both forms of bullying victimisation across 27 European countries: Albania, Austria, Belgium, Bulgaria, Czech Republic, Germany, Estonia, Spain, France, Croatia, Hungary, Ireland, Israel, Iceland, Italy, Luxembourg, Moldova, Malta, the Netherlands, Portugal, Romania, Russia, Ukraine, Sweden, Slovenia, Slovakia, and the UK.

The social–ecological model underscores the moderating role of different sources of perceived social support within multilayered social contexts [2]. Consistent with Bronfenbrenner’s [13] social–ecological model, empirical studies suggest that adolescents receive perceived social support from several sources (e.g., teachers and families) and social networks within their social environment on a regular basis [27,28]. Perceived social support refers to the extent to which individuals believe that they can receive support (i.e., information, assistance, and resources) from different sources in their social environment (i.e., families, peers, friends, and teachers, particularly when they need help), leading them to believe that they are cared for, esteemed or valued, loved, and belong to a large social network with mutual obligation [29,30,31]. Social support involves instrumental support (e.g., material assistance or financial aid), informational support (e.g., knowledge and skills), emotional support (e.g., empathy, caring, reassurance or encouragement, and trust), and appraisal social support, such as evaluative feedback or information [28,32,33,34].

Existing studies focusing on the moderating role of teacher support suggest that perceived teacher support provides a significant buffer for adolescents against the association between bullying perpetration and bullying victimisation [35,36,37]. High levels of social support from teachers (i.e., supportive school environments in which students perceive teachers as warm, caring, kind, trusted, and helpful and make adolescents feel emotionally safe) play a significant role in protecting adolescents from school bullying and victimisation [37]. Moreover, teacher support (such as close monitoring of students during school hours and direct interventions, including teachers’ responses and attitudes towards aggression, coping efficiency, and strategies) greatly influences adolescent bullying behaviours [38]. It is likely that teachers take appropriate measures to deal with bullying behaviours as opposed to peers, who may not consider bullying perpetration and victimisation as problematic life experiences among adolescents [39,40,41]. It is also possible that adolescents who learn in supportive environments with quality school and classroom climates that foster stronger connections (i.e., warm and trusting relationships) between teachers and students would be more likely to report violent and bullying behaviours to their teachers [39,40]. Furthermore, high levels of teacher social support may provide adolescents with psychological and informational support that might enhance their sense of belonging or school connectedness, which would, in turn, lead to reduced rates of bullying perpetration and the risk of victimisation [42,43]. Despite the fact that existing studies have evidenced the moderating role of teacher social support in the association between bullying perpetration and bullying victimisation [26], it is still unclear whether perceived teacher social support moderates the linkage between bullying perpetration and bullying victimisation at the cross-national level. Moreover, most of the studies focusing on the attenuating effects of perceived teacher social support on the association between bullying perpetration and victimisation were single-country analyses as opposed to multilevel cross-national analyses [16,44,45]. Thus far, cross-national research focusing on the moderating effects of perceived teacher social support on the association between traditional bullying perpetration and victimisation by traditional bullying and cyberbullying across European countries is still scant. Therefore, the present study sought to use the 2017/18 HBSC data to examine the attenuating effects of perceived teacher social support on the influences of traditional bullying perpetration on both forms of bullying victimisation across 27 countries in Europe.

Prior research suggests that perceived family social support offers protection for adolescents from bullying behaviours [2,46,47]. The extant literature provides explanations for the moderating effects of family support on the associations between adolescent risky behaviours and bullying victimisation, suggesting that not only do family members provide material and financial assistance but also emotional and informational support that enables adolescents to become resilient and cope with experiences of problems associated with aggression and peer victimisation, such as offline and cybervictimisation [48,49]. Additionally, evidence indicates that high levels of family social support, such as parental monitoring of adolescents, positive parental involvement in socialising children, good parenting styles, sibling support, positive adolescent–parent relations, and effective communication, are associated with reduced odds of bullying victimisation [2,46,47,50]. However, poor family support, including ineffective family communication, negative parent-adolescent relationships, poor relationships with family members, a lack of parental supervision, and poor parenting styles, may be related to increased odds of bullying victimisation [50,51]. Prior research affirms that lower levels of family social support might be associated with higher levels of adolescents’ involvement in delinquent lifestyle behaviours (e.g., bullying perpetration, physical fighting, alcohol use, smoking, and drug use) and greater risks of bullying victimisation [47,52,53,54]. Although the literature provides theoretical and empirical support for the moderating role of family social support in the association between bullying perpetration and bullying victimisation, cross-national research examining the moderating effects of perceived family social support on these associations is still limited. To date, mainly single-country studies have examined the moderating role of perceived family social support in the association between bullying perpetration and victimisation. Thus, there is a scarcity of multilevel cross-national studies that have examined the attenuating effects of perceived family social support on the association between traditional bullying perpetration and victimisation by cyberbullying and traditional bullying by using large nationally representative samples of adolescents from the 2017/18 Health Behaviour in School-aged Children (HBSC) survey data from 27 European countries.

Existing cross-national studies have utilised several proxies or indicators of macro-level social support (e.g., percent of GDP spent on public education and health care, quality of human development, decommodification index, and social protection) to examine the moderating effects of social support on the association of criminal behaviours with victimisation in Western countries [33,55,56]. These studies have found that macro-level social support acts as a buffer to prevent individuals from perpetrating criminal acts (such as physical fights and assaults) and protect people from any forms of victimisation, such as homicide and bullying victimisation [24,33]. That is, macro-level social support moderates the influence of deviant acts (such as physical fighting and bullying perpetration) on victimisation [33], offering credence to the assertion that macro-level social support has a direct and indirect influence on victimisation [33]. Accordingly, prior cross-national studies suggest that proxy measures of macro-level social support (e.g., public health care spending) promote and enhance well-being and social development among individuals, especially children [57,58,59]. However, up until now, most of the studies have focused on examining the moderating role of health care expenditure in the association between structural risk factors (e.g., income inequality) and violent victimisation (e.g., homicide). Existing literature indicates a dearth of cross-national studies that have examined the moderating effects of public health care spending on the association of traditional bullying perpetration with traditional bullying and cyberbullying victimisation [24]. Consequently, less is known about how public health care spending moderates the association between traditional bullying perpetration and victimisation by traditional bullying and cyberbullying across European countries. To that effect, multilevel cross-national research is needed to examine the moderating effects of public health care spending (i.e., proxy macro-level social support) on the association between traditional bullying perpetration and victimisation by traditional bullying and cyberbullying using data from the 2017/18 Health Behaviour in School-aged Children (HBSC) survey from 27 countries in Europe.

## 2. The Current Study

The social–ecological perspective holds that adolescents’ behaviour problems and well-being are not only products of their interactions with ecological environments but also their lifestyle choices within the microsystem, such as the peer, family, school, and community contexts [13]. Supporting theoretical notions, previous single-country studies have found that adolescents’ involvement in bullying perpetration places adolescents at a greater risk of bullying victimisation [26,50]. However, until now, very few cross-national studies have examined how traditional bullying perpetration predicts traditional bullying and cyberbullying victimisation across European countries [24,60]. Thus, less is known about how traditional bullying perpetration relates to traditional bullying and cyberbullying victimisation among adolescents across 27 countries in Europe.

An extant body of theoretical and empirical literature suggests that high levels of social support are associated with lower rates of victimisation [33,56]. For example, the social support theory posits that individuals who receive high social support are less likely to engage in risky lifestyles and deviant behaviours [33,56,61]. Moreover, the empirical literature indicates that distinct forms of social support have different impacts on deviant behaviours (e.g., bullying perpetration) and their association with victimisation [62]. Although theoretical and empirical literature provides support for the positive association between bullying perpetration and bullying victimisation [62,63,64,65], there is still a limited scientific understanding of how each type of social support (i.e., family, teacher, and health care support) moderates the association between traditional bullying perpetration and victimisation by traditional bullying and cyberbullying at the cross-national level. Thus, the moderating role of each dimension of social support in the relationship between bullying perpetration and victimisation is not yet well understood. To address this gap, the present study sought to examine how public health care spending and perceived social support from families and teachers moderate the linkage between traditional bullying perpetration and victimisation by traditional bullying and cyberbullying across 27 countries in Europe. The current study sought to address the following research questions:How does traditional bullying perpetration influence traditional bullying and cyberbullying victimisation? This study hypothesised that adolescents who perpetrate traditional bullying are more likely to be victims of traditional bullying and cyberbullying than those who do not bully others.How does each type of perceived social support (i.e., family and teacher support) buffer the main effect of traditional bullying perpetration on victimisation by traditional bullying and cyberbullying? This study hypothesised that adolescents who perceive high family and teacher social support are less likely to perpetrate traditional bullying and thus be at a lower risk of being victimised by traditional bullying and cyberbullying than those who perceive low social support.How does public health care support buffer the magnitude of the positive association between traditional bullying perpetration and victimisation by traditional bullying and cyberbullying at the cross-national level? This study hypothesised that high levels of public health care support would buffer adolescents against the positive association between traditional bullying perpetration and victimisation by traditional bullying and cyberbullying.

## 3. Materials and Methods

### 3.1. Data Sources and Study Design

A multilevel quantitative cross-national research approach was used to examine individual- and country-level data [66,67]. The current study combined country data with the 2017/18 Health Behaviour in School-aged Children (HBSC) Survey data. Self-reported HBSC data were collected from nationally representative samples of 11-year-old, 13-year-old, and 15-year-old adolescents using a standardised research protocol in 47 European and North American countries [68]. Two-stage cluster sampling was used to select schools and then adolescents from school classes in each country. Each HBSC participating country subjected the survey procedures and standardised self-report questionnaires to a research ethics review. The analytic sample was based on 162,792 adolescents (80,181: 49.3% boys and 82,611: 50.7% girls; mean age of 13.49 years; standard deviation = 1.62). This study includes data from 27 countries (i.e., countries that had complete cases regarding key individual- and country-level variables for the multilevel analyses).

### 3.2. Measures

#### 3.2.1. Dependent Variables

*Traditional bullying victimisation* was measured on a 5-point Likert scale using a single item [68]. The adolescents were asked, “How often have you been bullied at school in the past couple of months?” The responses to the item ranged between 1 = “I have not been bullied at school in the past couple of months”, 2 = “It has only happened once or twice”, 3 = “2 or 3 times a month”, 4 = “about once a week”, and 5 = “several times a week”. Responses 1 and 2 were scored as 0 (not been bullied), and responses 3 to 5 were recoded as 1 (been bullied), in line with previous studies [69,70,71]. *Cyberbullying victimisation* was also measured using a single item on a 5-point Likert scale [68]. The adolescents were asked: “How often have you been cyberbullied (i.e., someone sent mean instant messages, email, or text messages about you; wall postings; created a website making fun of you; posted unflattering or inappropriate pictures of you online without permission or shared them with others in the past couple of months)?” The responses ranged between 1 = “have not”, 2 = “once or twice”, 3 = “2 or 3 times per month”, 4 = “once a week”, and 5 = “several times a week”. Responses 1 and 2 were recoded as 0 (not been cyberbullied), and responses 3 to 5 were scored as 1 (been cyberbullied), in line with previous studies [41,71,72].

#### 3.2.2. Individual-Level Independent Variable(s)

*Traditional bullying perpetration* was used to measure the element of risky behaviours among adolescents. Based on the HBSC 2017/18 survey, traditional bullying perpetration was assessed using a single question based on a 5-point scale. The adolescents were asked, “How often have you taken part in bullying another person/student(s) at school in the past couple of months?” The responses ranged from 1 = “I have not bullied another person/student(s)” to 5 = “several times a week” [68]. In line with earlier research [24,73], all the responses were dichotomised, resulting in a dummy variable (0 = “not bullied another student”; 1 = “bullied another student”).

#### 3.2.3. Individual-Level Social Support (Perceived Social Support)

*Family social support* was measured using four items (my family really tries to help me; I receive the emotional help and support I need from my family; I can talk about my problems with my family; and my family is willing to help me make decisions; α = 0.94) based on the subscale of the Multidimensional Scale of Perceived Social Support [68]. All four items were measured on a 7-point Likert scale, with response choices ranging from “1 = very strongly disagree” to “7 = very strongly agree”. In the present study, values from the four items for the subscale were computed to obtain a composite score, with low scores indicating low social support and high scores denoting high social support.

*Teacher social support* was measured by three items (I feel that my teachers accept me as I am; I feel that my teachers care about me as a person; and I feel a lot of trust in my teachers; α = 0.83). The response options were captured on a 5-point Likert scale as follows: “5 = strongly agree, 4 = agree, 3 = neither agree nor disagree, 2 = disagree, and 1 = strongly disagree”. In the present study, the response options were reverse coded (i.e., changed from a positively ordered to a negatively ordered pattern) and ranged between “1 = strongly disagree, 2 = disagree, 3 = neither agree nor disagree, 4 = agree, and 5 = strongly agree” (i.e., 1 = low social support to 5 = high social support). Then, all the values from the three items were computed to obtain a composite score, with low scores indicating low social support and high scores indicating high social support. The scales have been validated and used to conduct research using various samples involving adolescents [68,74].

#### 3.2.4. Individual-Level Control Variables

*Age* was measured as a categorical variable (11-year-olds, 13-year-olds, and 15-year-olds). The *gender* variable was measured as a dummy variable (0 = male, 1 = female) based on the 2017–2018 HBSC scale [68]. *Family affluence* was assessed using six items measuring family affluence (i.e., cars, computers, bedrooms, bathrooms, dishwashers, and vacation/holiday) based on the HBSC family affluence (FAS) scale [68]. All the values from the six items were recoded and computed to obtain an index variable (sum score), with the high values indicating the most affluent and the low values indicating the least affluent [69,75,76].

#### 3.2.5. Country-Level Social Support

*Public health care spending (i.e., health care support)* was measured as a proxy for country-level social support based on a proportion of the gross domestic product that each country invests in the provision of public health care [77,78,79]. The indicator is a proxy measure of macro-level social support since investment in health care signifies measures of “decommodification” [80]. The data were obtained from OECD databases [81].

#### 3.2.6. Country-Level Control Variable(s)

This study used *gross domestic product (GDP) per capita* expressed in U.S. dollars using purchasing power parity rates [78,82,83]. GDP per capita measures were obtained from the United Nations Development Programme database [84].

### 3.3. Statistical Analyses

This study used SPSS version 29 to conduct the statistical analyses. The present study determined whether multicollinearity between independent variables existed. The condition index scores did not exceed 30, and none of the zero-order correlations exceeded 0.70. Additionally, variance inflation factors (VIFs) did not exceed the values of 5 to 10, and the overall values of tolerance were not less than the range of 0.1 to 0.2. As a result, multicollinearity was not a major problem [85,86,87]. Since the present study did not simultaneously test multiple hypotheses, multiplicity in the regression analyses was not a major problem [88,89,90]. This study used separate models to test the hypothesis of the association between traditional bullying perpetration and each form of bullying victimisation using a significance level of 0.05, adjusting for covariates. This study also used separate regression models to test the moderating effects of each type of perceived social support and health care spending on the association between traditional bullying perpetration and each form of bullying victimisation, with an alpha of 0.05.

This study conducted descriptive analyses for sample characteristics and country-level variables. Taking into account the hierarchical nesting of adolescents within schools and the clustering of schools within countries, three-level regression models were estimated [66,91,92]. Specifically, multilevel mixed-effects binary logistic regression models were performed since both outcome variables were dummy variables [24,93]. Five models were estimated for each outcome variable. First, Model 1 (the null model) estimated the extent of variability of each outcome variable (i.e., traditional bullying and cyberbullying victimisation) across schools (level 2) and countries (level 3). Second, Model 2 examined the association between traditional bullying perpetration and each form of bullying victimisation across countries, adjusting for covariates (i.e., age, family affluence, and gender). Third, Model 3 included the main effect of family social support and within-level interactions to examine the moderating effects of family social support on the association between traditional bullying perpetration and each form of bullying victimisation. Fourth, Model 4 included the main effect of teacher social support and within-level interactions to examine the moderating effects of teacher social support on the association between traditional bullying perpetration and each form of bullying victimisation. Finally, Model 5 included the main effect of public health care spending and cross-level interactions to examine the moderating effects of public health care spending on the association between traditional bullying perpetration and each form of bullying victimisation, controlling for the effects of gross domestic product (GDP) per capita. The present study employed SPSS PROCESS Macro 4.2 [94] to examine the moderating role of perceived social support and health care spending in the association between traditional bullying perpetration and victimisation by traditional bullying and cyberbullying. This study also conducted slope analyses to interpret moderation effects. Despite the fact that this study did not test any hypotheses at level 2 (school level), the school level was specified in the regression models to account for a design effect of the school cluster [24,67,95].

## 4. Results

### 4.1. Descriptive Analyses of Individual and Country-Level Sample Characteristics

The descriptive statistics on key variables are shown in Table 1. The proportions of girls ranged from 48.7% (Slovakia) to 54.8% (Israel), while the proportions of boys ranged from 45.2% (Israel) to 51.3% (Slovakia). The mean age was 13.49 years (standard deviation = 1.63). The prevalence of victimisation by traditional bullying was 28.8%, whereas the frequency of cyberbullying victimisation was 16.9% for the whole sample. Likewise, the prevalence of traditional bullying perpetration was 22.1% for the total sample.

The country-level characteristics (i.e., GDP per capita and public health expenditure) varied across countries. Gross domestic product (GDP) per capita ranged from USD 5190 (Republic of Moldova) to USD 94,278 (Luxembourg). Regarding public spending on health care, Table 1 shows that the percentage of GDP expenditure on health care ranged from 5.2% (Albania) to 11.5% (Germany). The average score for gross domestic product (GDP) per capita was 35,552.01 (standard deviation = 15,050.50), and for public health care spending, it was 8.47 (standard deviation = 1.77).

### 4.2. Relationship between Traditional Bullying Perpetration and Victimisation by Traditional Bullying and Cyberbullying

This study examined whether adolescents who engage in traditional bullying perpetration are more likely to be victims of traditional bullying and cyberbullying compared with those who do not perpetrate bullying, controlling for the effects of individual-level covariates (age, family affluence, and gender), as shown in Table 2 and Table 3 below. The coefficients (β = 0.27, *p* < 0.001; β = 0.25, *p* < 0.001) in the null models in Table 2 and Table 3 indicated that the intercept variance varied across schools, respectively. The results suggest a statistically significant variance in the probability of being victimised by traditional bullying and cyberbullying among adolescents who bully others across schools. The intraclass correlation (ICC) showed that about 5% of traditional bullying victimisation variance was explained across schools (as can be seen in the null model 1 in Table 2), whereas about 3% of cyberbullying victimisation variance was explained across schools (as shown in the null model 1 in Table 3). These results suggest that about 5% and 3% of the variability in traditional bullying and cyberbullying victimisation lies between schools, respectively. Additionally, the coefficients (β = 0.10, *p* < 0.001; β = 0.12, *p* < 0.001) in the null models in Table 2 and Table 3 showed that the intercept variance varied between countries, respectively. This, therefore, suggests a statistically significant variance in the probability of being victimised by traditional bullying and cyberbullying across countries. The ICC indicated that about 2% of the traditional bullying victimisation variance was explained across countries. Similarly, around 2% of the cyberbullying victimisation variance was explained at the cross-national level. These results suggest that about 2% of the variability in traditional bullying and cyberbullying victimisation lies between countries, respectively.

Supporting the research hypothesis, Model 2 in Table 2 showed that traditional bullying perpetration significantly predicted traditional bullying victimisation at the cross-national level (OR = 5.32; 95% CI = 5.18, 5.46; *p* < 0.001), controlling for individual-level covariates (age, family affluence, and gender). The results showed that traditional bullying perpetration multiplies by 5.32 times the probability of being bullied among perpetrators of bullying compared to non-perpetrators. These findings suggest that each standard deviation increase in rates of traditional bullying perpetration corresponded with an increase in the odds of traditional bullying victimisation. The results demonstrate that there is a higher risk of being victimised by traditional bullying among adolescents who perpetrate bullying than those who do not engage in traditional bullying perpetration. Accordingly, the results suggest that traditional bullying perpetration is a significant predictor of traditional bullying victimisation among adolescents across countries. Additionally, Model 2 in Table 3 indicated that there was a positive and statistically significant relationship between traditional bullying perpetration and cyberbullying victimisation (OR = 5.14; 95% CI = 5.00, 5.30; *p* < 0.001), controlling for the effects of individual-level covariates. These results suggest that traditional bullying perpetration is strongly associated with a high risk of cyberbullying victimisation at the cross-national level. The odds ratio suggests that adolescents who are involved in traditional bullying perpetration have 5.14 times more odds of being victimised by cyberbullying than those who are not involved in bullying perpetration. These results suggest that a one-standard-deviation increase in traditional bullying perpetration was largely associated with an increase in the likelihood of cyberbullying victimisation. Hence, the results suggest that adolescents who bully others are more likely to be at high risk of cyberbullying victimisation than those who do not bully others.

The results also revealed that the main effect of traditional bullying perpetration on traditional bullying victimisation was still positive and statistically significant (OR = 7.25; 95% CI = 6.87–7.64; *p* < 0.001) and (OR = 6.97; 95% CI = 6.70–7.25; *p* < 0.001) even after the inclusion of individual-level moderators and interactions in the regression models, as shown in models 3 and 4 in Table 2. Models 3 and 4 in Table 3 showed that the main effect of traditional bullying perpetration on cyberbullying victimisation was still positive and statistically significant (OR = 7.85; 95% CI = 7.44–8.28; *p* < 0.001) and (OR = 7.77; 95% CI = 7.44–8.11; *p* < 0.001) after the inclusion of individual-level moderators and interactions. Moreover, the results showed that the main effect of traditional bullying perpetration on victimisation by traditional bullying (OR = 5.26; 95% CI = 5.05, 5.48; *p* < 0.001) and cyberbullying (OR = 5.64; 95% CI = 5.40–5.90; *p* < 0.001) was still positive and statistically significant after the country-level variables and interactions were included in the regression models, as can be seen in Model 5 in each table.

This study also examined whether family social support moderates the association of traditional bullying perpetration with traditional bullying and cyberbullying victimisation among adolescents across countries. The results showed that the interaction of traditional bullying perpetration and family social support was negative and statistically significant, suggesting that family social support moderated the association between traditional bullying perpetration and victimisation by traditional bullying (OR = 0.62; 95% CI = 0.58–0.66; *p* < 0.001) and cyberbullying (OR = 0.48; 95% CI = 0.45–0.51; *p* < 0.001). As shown in Figure 1 and Figure 2 below, the slopes of the lines indicated that the association between traditional bullying perpetration and victimisation by traditional bullying and cyberbullying was significantly weaker for adolescents who reported higher family social support (1 SD above the mean) compared to those who reported lower family social support (1 SD below the mean), supporting the study hypothesis. The results in Figure 1 and Figure 2 show that the top blue lines for adolescents who reported lower family social support are much steeper than the bottom red lines for those who reported higher family social support, indicating the stronger effect of traditional bullying perpetration on traditional bullying and cyberbullying victimisation for adolescents with lower family social support.

Furthermore, this study examined whether teacher social support moderates the association of traditional bullying perpetration with traditional bullying and cyberbullying victimisation. The results showed that teacher social support moderated the association between traditional bullying perpetration and victimisation by traditional bullying (OR = 0.61; 95% CI = 0.58–0.64; *p* < 0.001) and cyberbullying (OR = 0.47; 95% CI = 0.45–0.50; *p* < 0.001). As can be seen in Figure 3 and Figure 4 below, the slopes of the lines indicated that the association of traditional bullying perpetration with traditional bullying and cyberbullying victimisation was significantly weaker for adolescents who reported more teacher social support (1 SD above the mean) compared to adolescents who reported less teacher social support (1 SD below the mean), confirming the study hypothesis. Figure 3 and Figure 4 show that the top blue lines representing adolescents with less teacher social support are much steeper than the bottom red lines for those with more teacher social support, indicating the stronger effect of traditional bullying perpetration on traditional bullying and cyberbullying victimisation for adolescents with less teacher social support.

Furthermore, this study sought to examine how public health care spending would moderate the association of traditional bullying perpetration with traditional bullying and cyberbullying victimisation among adolescents at a cross-national level. The results indicated that health care spending moderated the association of traditional bullying perpetration with victimisation by traditional bullying (OR = 0.93; 95% CI = 0.88–0.98; *p* < 0.01) and cyberbullying (OR = 0.74; 95% CI = 0.69–0.78; *p* < 0.001). As can be seen in Figure 5 and Figure 6, the slopes of the lines indicated that the relationship between traditional bullying perpetration and victimisation by traditional bullying and cyberbullying was weaker for adolescents with higher health care support (1 SD above the mean) compared to adolescents with lower health care support (1 SD below the mean), supporting the study hypothesis. The results in Figure 5 and Figure 6 show that the top blue lines for low health care support are much steeper than the bottom red lines for high health care support, indicating the stronger effect of traditional bullying perpetration on traditional bullying and cyberbullying victimisation among adolescents with lower health care support.

## 5. Discussion

The present cross-national HBSC study examined how traditional bullying perpetration influences victimisation by traditional bullying and cyberbullying across 27 European countries. This study found that traditional bullying perpetration was significantly associated with traditional bullying and cyberbullying victimisation among adolescents. This study expands the existing literature on the main effect of bullying perpetration on bullying victimisation by indicating that greater rates of traditional bullying perpetration are associated with higher odds of traditional bullying and cyberbullying victimisation [24,50]. Supporting the research hypothesis and consistent with existing findings [20,96], the findings of this study suggest that adolescents who frequently perpetrate bullying are more likely to be victims of traditional bullying and cyberbullying compared to those who do not bully others within social environments, such as peer and school contexts. These findings are echoed by findings from prior research showing that bullying others is an adolescent high-risk behaviour that exposes adolescents to a high risk of bullying victimisation [25]. Specifically, the current findings are aligned with the findings of a recent HBSC cross-national study [24], in which traditional bullying perpetration was found to be a substantial risk factor that puts adolescents at high risk for traditional bullying and cyberbullying victimisation. The findings of this study provide empirical support for the social–ecological perspective [26,50] by suggesting that adolescents’ involvement in bullying perpetration creates a situational context for repeated aggression and bullying victimisation within the peer and school contexts [16,70,97,98]. It is likely that adolescents’ engagement in bullying perpetration exposes them to more powerful bullies in risky environments in which societal norms condone or model violence and peer norms support peer conflicts and aggressive behaviours [98,99,100] Moreover, it is plausible that adolescents who lack social support might frequently engage in bullying others and subsequently be at high risk of bullying victimisation, especially since bullying often occurs away from adults, such as parents and teachers, who can offer some protection [17,101]. As suggested by Azimi and Daigle [62], the absence of social support would not only be a precipitating factor that would make the youth indulge in risky behaviours (such as bullying perpetration) but also expose adolescents to higher risks of peer victimisation.

The present study also examined whether each type of perceived social support (i.e., family and teacher support) moderates the association between traditional bullying perpetration and victimisation by traditional bullying and cyberbullying among adolescents across 27 European countries. Generally, supporting the research hypothesis, both sources of perceived social support (i.e., family and teacher support) consistently moderated the association between traditional bullying perpetration and victimisation by traditional and cyberbullying victimisation, such that the association was significantly weaker for adolescents with higher social support than those with lower social support. This study provides evidence in support of the microsystem of the social–ecological model on multidimensional perceived social support [97,102], suggesting that multilayered systems of social support can serve as a significant buffer for adolescents against bullying perpetration and bullying victimisation.

Regarding the moderating effects of family social support on the association between traditional bullying perpetration and victimisation by traditional bullying and cyberbullying, this study found that the risk of being victimised by cyberbullying and traditional bullying due to involvement in bullying perpetration was significantly mitigated among adolescents who reported more social support from families. In line with findings from past research [2,47,70], this study demonstrates that higher family social support is associated with lower odds of bullying perpetration and victimisation. The current study makes a key contribution to the empirical literature [103] by suggesting that family social support is a substantial protective factor for adolescents against the positive association between traditional bullying perpetration and victimisation by traditional bullying and cyberbullying. Moreover, the findings of this study provide empirical support for the microsystem of the social–ecological model [2,104], indicating that family social support offers significant attenuating effects on bullying behaviours among adolescents within social–ecological environments, such as peer and school contexts. As posited by the social–ecological model [2,96,105], the current findings highlight that family social support is a contextual protective factor that interacts with risky behaviours (such as bullying behaviours) to influence the well-being of adolescents within the child’s social ecology. Generally, in line with past research findings [27,106,107], the current findings suggest that adolescents with higher perceived family social support are less likely to engage in traditional bullying perpetration and thus be at a lower risk of being victimised by traditional bullying and cyberbullying. These findings suggest that family social support not only serves as a significant protection for adolescents from indulging in risky behaviours (including bullying perpetration) but also buffers adolescents against risks of peer victimisation (such as traditional and cyberbullying victimisation). To that effect, the current findings offer support for the existing literature suggesting that family social support is an important source of strength for adolescents and has been associated with positive adolescent outcomes (e.g., improved well-being), which can buffer adolescents against bullying victimisation [108]. It is likely that adequate family social support (e.g., emotional and informational support) promotes prosocial behaviours, a sense of belonging, mutual respect, and connectedness of adolescents to families, peers, and schools, which, in turn, lead to reduced rates of bullying perpetration and victimisation among adolescents [106,109]. Additionally, it is possible that family support (e.g., financial assistance, material, emotional, and informational support provided by family members) might enable adolescents to develop coping skills to deal with aggression and peer victimisation [49,110]. Moreover, high levels of family support (such as strong family connections with adolescents, a caring home environment, effective communication with adolescents, parental monitoring, and good parenting styles) enable parents to directly intervene and prevent adolescents from perpetrating bullying, as well as protect adolescents from bullying victimisation [50,103].

In line with the findings from previous research [37], the present study found that high-quality social support from teachers (i.e., supportive school environments in which students perceive teachers as warm, kind, trusted, and helpful and make adolescents feel emotionally safe) offered significant protection for adolescents from bullying perpetration and victimisation. Thus, one vital contribution of this cross-national study is that it extends existing literature [39,40] by suggesting that teacher social support served as a contextual protective factor that buffered adolescents against the main effect of traditional bullying perpetration on traditional bullying and cyberbullying victimisation. The present findings point to the important role of teacher social support in adolescent development and socialisation in the school context, especially since adolescents spend most of their time with teachers in school, away from primary caregivers [37]. It is plausible that teachers take appropriate measures to deal with bullying behaviours as opposed to peers who may not consider bullying perpetration and victimisation as problematic life experiences among adolescents [39,40,111]. Additionally, it is possible that adolescents who learn in supportive environments with quality school and classroom climates that foster stronger connections (i.e., warm and trusting relationships) between teachers and students would be more likely to report violent and bullying behaviours to their teachers [39,40,111]. Moreover, higher levels of teacher social support may provide adolescents with psychological and informational support that might enhance their sense of belonging or school connectedness, which would, in turn, lead to reduced rates of bullying perpetration and the risk of victimisation [42,43]. Primarily, this study found that adolescents who reported higher teacher social support were less likely to engage in bullying perpetration and hence be at lower risk of victimisation by both traditional and cyberbullying. Conversely, adolescents who reported lower teacher support were more likely to report bullying others and being bullied. These findings are consistent with past research findings [39], indicating that teacher social support (i.e., a caring school environment in which teachers provide quality support, guardianship, and close monitoring of students during school hours) leads to decreased adolescent bullying perpetration and reduced risk of bullying victimisation. As echoed by Veenstra and associates [38], teacher support (i.e., direct interventions, including teachers’ response and attitudes towards aggression, coping efficiency, and strategies) prevents adolescents from bullying others as well as protects them from being bullied. The findings of the present study offer support for the social–ecological framework [112], suggesting that teachers are considered to play a fundamental social role in the detection and prevention of bullying perpetration and the protection of victims within the school context. Moreover, teachers play an important role in the implementation of many anti-bullying programmes aimed at reducing bullying since they are able to identify and respond to bullying situations within the school context [113,114,115]. Thus, it is possible that school anti-bullying policies, prevention, and intervention strategies that enhance teacher social support measures (such as teacher resources) lead to decreased rates of bullying perpetration and reduced odds of victimisation by traditional bullying and cyberbullying [116,117]. For example, evidence from existing research [54,114,118,119] has confirmed that effective implementation of school anti-bullying interventions has resulted in significant reductions in traditional bully-victim and cyberbully-victim rates across Western European countries (such as England, Wales, Norway, Sweden, Belgium, Germany, the Netherlands, Finland, Spain, Switzerland, Ireland, and Austria). However, future studies may be needed to examine the influence of school-based prevention and intervention programmes on the relationship between adolescent risky behaviours (e.g., bullying perpetration) and bullying victimisation.

This study also tested the hypothesis of whether public health care expenditure (i.e., proxy country-level social support) would moderate the association of traditional bullying perpetration with victimisation by traditional bullying and cyberbullying among adolescents across 27 European countries. Consistent with the study hypothesis, the findings showed that health care spending attenuated the association between traditional bullying perpetration and victimisation by traditional bullying and cyberbullying. Accordingly, the findings of the present study make a significant contribution to the literature by demonstrating that high public health care support offered a strong buffer for adolescents against the main effect of traditional bullying perpetration on cyberbullying and traditional bullying victimisation [33,55]. These results suggest that high health care support not only prevents adolescents from perpetrating traditional bullying but also reduces the risk of victimisation by cyberbullying and traditional bullying. The findings of this cross-national study are consistent with existing empirical notions suggesting that macro-level social support can indirectly influence victimisation by buffering or cushioning the effects of risk factors that predispose people to victimisation [33,83]. This study provides support for the macrosystem of the social–ecological model by showing that health care support served as a significant macro-level protective factor for adolescents against the association between traditional bullying perpetration and victimisation by traditional bullying and cyberbullying among adolescents in 27 European countries [33,56]. One possible explanation for the buffering effects of health care support on the association of traditional bullying perpetration with traditional bullying and cyberbullying victimisation is that nations that provide not only higher levels of social support but also appropriate sources of social support have lower rates of bullying perpetration and victimisation [33,56]. Another possible explanation is that huge investment in public health programmes might provide support for anti-bullying prevention and intervention strategies to address bullying behaviours and stressful situations associated with victimisation among adolescents in schools across countries. Moreover, it is possible that high public health expenditures may be targeted at devising normative re-educative strategies to re-educate adolescents on risks for bullying behaviours and address societal norms or peer group norms that condone competition, power differences, social dominance, social hierarchies, conflict, aggressive behaviours, and violence in society. In the end, it is possible that public health education would help societies by promoting and embracing shared values and beliefs that foster social cohesion and social support for everyone, as well as promoting social control for aggressive behaviours and victimisation of other people [120,121]. Accordingly, the association of traditional bullying perpetration with traditional bullying and cyberbullying victimisation would be buffered in countries with high levels of health care support. This implies that substantial investment in health care programmes on bullying prevention would serve as a buffer against bullying perpetration and victimisation [33,56]. Existing empirical evidence suggests that social support might provide buffering effects that would act as protective and coping mechanisms to reduce the likelihood of victimisation [33,122]. Additionally, it is possible that the provision of quality public health programmes on bullying intervention might cushion the negative effects of bullying perpetration and victimisation among adolescents, which might, in turn, enhance the quality of life and well-being of adolescents in society [33].

### 5.1. Strengths and Limitations of the Study

The current study has the following strengths: First, this study combined country data with 2017/18 Health Behaviour in School-aged Children (HBSC) survey data. Second, this study utilised large nationally representative samples of adolescents and conducted mixed-effects multilevel analyses by taking into account the hierarchical clustering of adolescents in schools and the nesting of schools across countries [66]. Third, this study innovatively and simultaneously examined the moderating role of perceived social support and health care spending in the association between traditional bullying perpetration and victimisation by traditional bullying and cyberbullying. Finally, unlike previous studies that used ordinary research methods (i.e., single-level regression analyses), the current study employed a quantitative multilevel cross-national research approach to address methodological lacunae in prior studies [27,50,123].

Despite these strengths, this study has a number of limitations. First, this study did not establish cause-and-effect relationships as it used cross-sectional data from the 2017/18 Health Behaviour in School-aged Children (HBSC) survey [24,60]. Hence, future longitudinal studies might be appropriate to offer causal interpretations. Second, the HBSC surveys only use the self-completion questionnaire to collect data from adolescents. However, self-reported measures may not provide a clear picture of bullying behaviours [124]. Therefore, future studies should also use teacher- and parental-report questionnaires to collect data in order to provide a clear picture of bullying behaviours in adolescence. Third, based on the HBSC 2017–2018 survey, it is very difficult to distinguish between traditional bullying and cyberbullying as the definition is not comprehensive. That is, the measures of traditional bullying and cyberbullying are not adequately provided in the HBSC self-completion questionnaires [24,125]. Accordingly, it is difficult to obtain adequate indicators regarding adolescents’ experiences of bullying victimisation. Further research is therefore needed to use comprehensive measures or indicators of traditional bullying and cyberbullying in order to ascertain the nature of this phenomenon. Fourth, since the current study combined datasets from 27 countries in Europe, the results may not be generalised to other countries that have distinct socio–cultural and economic backgrounds [60]. To that effect, further research would be needed to replicate these findings in distinct socio–cultural and economic backgrounds outside Europe, especially in less developed countries. Fifth, the present study focused on examining the influence of microsystem (i.e., family and teacher social support) and macrosystem (health care support) ecological factors on the association between traditional bullying perpetration and victimisation by traditional bullying and cyberbullying. However, this study did not focus on examining how individual characteristics (such as socioeconomic status and gender) and ecological factors (such as community and cultural factors) influence the association between bullying perpetration and bullying victimisation [2]. To inform bullying prevention and intervention efforts, future research should examine how these individual characteristics and community and cultural factors influence the association between traditional bullying perpetration and bullying victimisation [2,26]. Finally, the present study employed secondary data from the 2017/18 HBSC survey. This entails that many years have passed since the data were collected. Hence, some changes in adolescent bullying behaviours and socioeconomic conditions might have taken place over such a period of time. For this reason, future research would be appropriate to examine trends in socioeconomic differences in bullying perpetration and bullying victimisation among adolescents.

### 5.2. Implications for Policy and Practice

The key finding of this study was that traditional bullying perpetration is risky behaviour that predisposes adolescents to a greater risk of traditional bullying and cyberbullying victimisation within social environments, such as the school context. This finding provides implications for bullying prevention and intervention efforts. To address adolescent bullying, it is imperative for policymakers to ensure that prevention and intervention efforts for bullying perpetration and victimisation utilise evidence-based strategies and services that work in concert [126]. In addition, it is important for practitioners (such as social workers, child psychologists, and child protection and welfare experts) to use the information and evidence from the findings of this study to advocate for policies that should have a strong focus on educating and sensitising adolescents, parents, community members, teachers, and other school staff on the risks of adolescent bullying perpetration and victimisation within the school and cyber contexts. Additionally, it is important for practitioners to advocate for policy innovations that would promote adolescent health, quality of life, and well-being not only among the victims of bullying but also among other youth across countries.

The current findings on the moderating role of perceived social support (i.e., family and teacher support) and health care support highlight the importance of promoting family, school, and health care systems of support (i.e., multilayered systems of support) in bullying prevention and intervention efforts. Accordingly, it is important to devise comprehensive prevention and intervention programmes that could reduce the risk of bullying victimisation by protecting adolescents from perpetrating bullying. Evidence-based intervention programmes (such as sensitisation and mentorship) should be designed to support efforts (such as counselling or emotional support) from social workers, psychologists, psychiatrists, parents, teachers, and other school staff with a view to enhancing emotional, interpersonal, and coping skills of adolescents (both victims of bullying and non-victims). Generally, multilayered policy interventions (e.g., case management, with key components on awareness raising, monitoring of adolescents, identification of bullying behaviours, reporting mechanisms, and support for adolescents) should be designed to foster quality social support from families, teachers, and health care institutions [33,127].

## 6. Conclusions

The findings suggest that traditional bullying perpetration is a substantial risk factor that predisposes adolescents to a high risk of traditional bullying and cyberbullying victimisation. These findings signify that adolescents who bully others offline and online are more likely to be victims of traditional bullying and cyberbullying compared with those who do not bully others. Additionally, the findings suggest that health care spending and perceived family and teacher social support appeared to play a significant moderating role in the association between the perpetration of traditional bullying and victimisation by traditional bullying and cyberbullying. The findings demonstrate that multilayered systems of social support (i.e., both micro- and macro-level social support) can deter adolescents from perpetrating traditional bullying, which can also reduce the risk of bullying victimisation. To protect adolescents from perpetrating traditional bullying that places them at high risk of bullying victimisation, multilayered systems of social support should be fully integrated into bullying prevention and intervention efforts.

## Figures and Tables

**Figure 1 ijerph-21-00863-f001:**
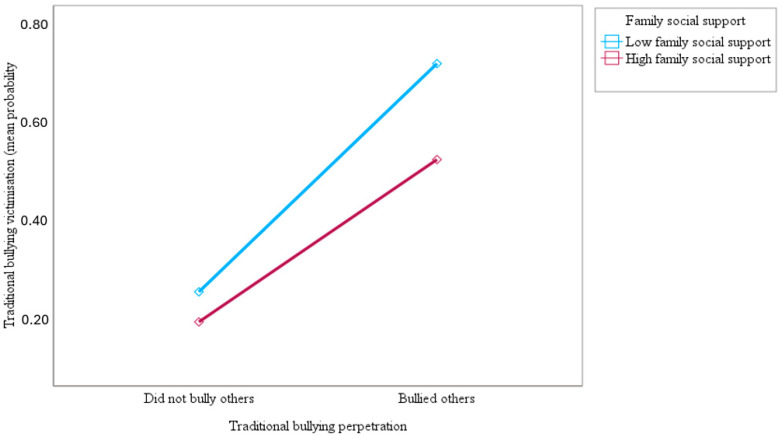
Slopes for the moderated association of traditional bullying perpetration with a predicted mean probability of traditional bullying victimisation at low and high (mean ± 1 SD) family social support.

**Figure 2 ijerph-21-00863-f002:**
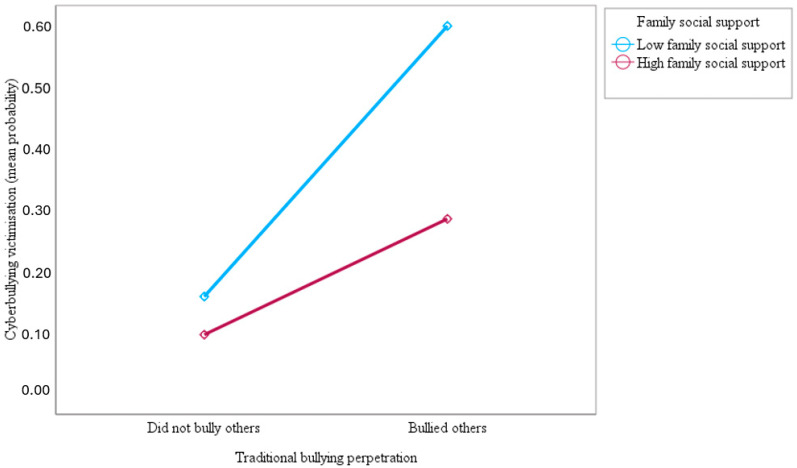
Slopes for the moderated association of traditional bullying perpetration with a predicted mean probability of cyberbullying victimisation at low and high (mean ± 1 SD) family social support.

**Figure 3 ijerph-21-00863-f003:**
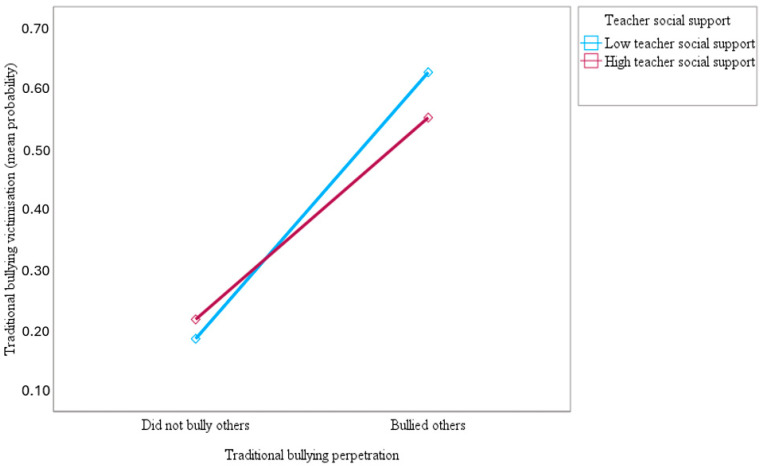
Slopes for the moderated association of traditional bullying perpetration with a predicted mean probability of traditional bullying victimisation at low and high (mean ± 1 SD) teacher social support.

**Figure 4 ijerph-21-00863-f004:**
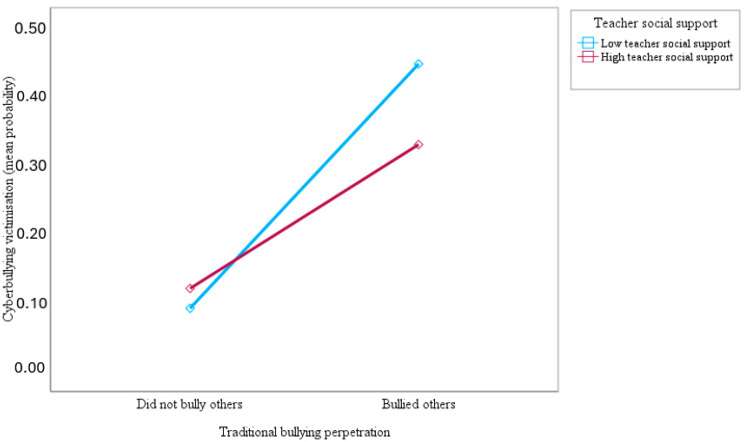
Slopes for the moderated association of traditional bullying perpetration with a predicted mean probability of cyberbullying victimisation at low and high (mean ± 1 SD) teacher social support.

**Figure 5 ijerph-21-00863-f005:**
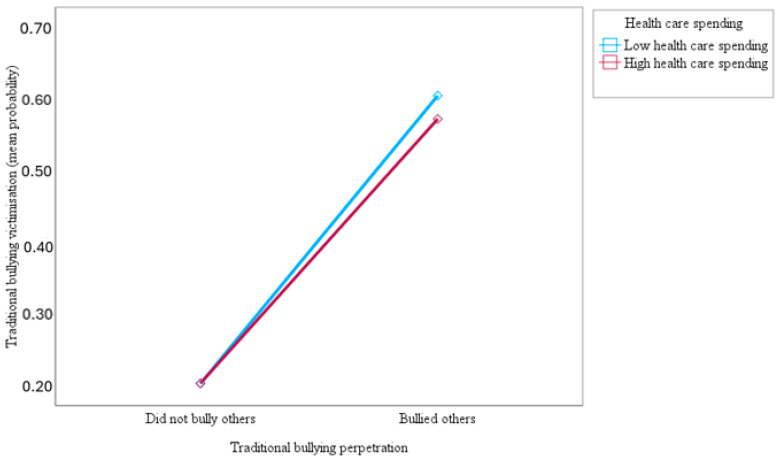
Slopes for the moderated association of traditional bullying perpetration with a predicted mean probability of traditional bullying victimisation at low and high (mean ± 1 SD) health care spending.

**Figure 6 ijerph-21-00863-f006:**
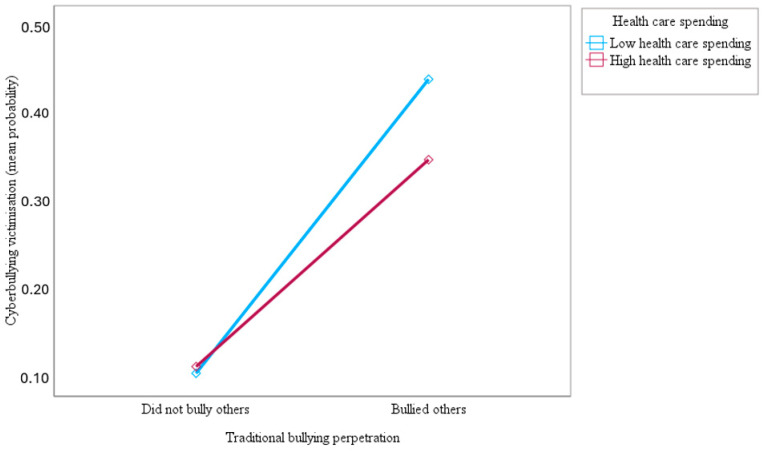
Slopes for the moderated association of traditional bullying perpetration with a predicted mean probability of cyberbullying victimisation at low and high (mean ± 1 SD) health care spending.

**Table 1 ijerph-21-00863-t001:** Descriptive statistics on key variables.

	Individual-Level Characteristics							Country-Level Characteristics	
		Gender		Age	Traditional Bullying Victimisation	Cyberbullying Victimisation	Traditional Bullying Perpetration	GDP per Capita	Health Care Spending
Country	*n*	Girls (%)	Boys (%)	Mean Age	*n* (%)	*n* (%)	*n* (%)	(USD)	% of GDP
Albania	1765	54.5	45.5	13.5	478 (11.1)	244 (5.6)	502 (28.4)	11,803	5.20
Austria	4129	50.7	49.3	13.3	579 (14.0)	368 (7.8)	735 (17.8)	45,437	10.30
Belgium ^a^	9911	50.2	49.8	13.3	679 (14.5)	381 (9.2)	1532 (15.5)	42,659	10.80
Bulgaria	4548	51.6	48.4	13.5	1182 (16.9)	406 (9.8)	1469 (32.3)	18,563	7.30
Czech Republic	11,564	49.7	50.3	13.4	1777 (19.4)	852 (11.0)	1715 (14.8)	32,606	7.50
Germany	4347	53.0	47.0	13.4	2336 (20.2)	1032 (11.3)	867 (19.9)	45,229	11.50
Estonia	4725	49.9	50.1	13.8	939 (22.4)	830 (11.9)	1338 (28.3)	29,481	6.70
Spain	4320	51.7	48.3	13.6	1436 (23.4)	1551 (13.4)	523 (12.1)	34,272	9.00
France	9170	50.5	49.5	13.3	980 (23.7)	821 (13.4)	1311 (14.3)	38,606	11.20
Croatia	5169	49.0	51.0	13.8	1030 (23.7)	589 (13.5)	1484 (28.7)	22,670	6.80
Hungary	3789	52.8	47.2	13.5	420 (23.8)	554 (13.6)	1212 (32)	26,778	6.60
Ireland	3833	49.4	50.6	13.4	1910 (24.8)	1361 (13.7)	625 (16.3)	67,335	6.90
Israel	7712	54.8	45.2	13.6	2480 (25.0)	252 (14.3)	1615 (20.9)	33,132	7.30
Iceland	6996	49.8	50.2	13.6	1084 (26.6)	860 (15.2)	663 (9.5)	46,483	8.40
Italy	4144	51.8	48.2	13.7	1545 (27.3)	412 (16.0)	544 (13.1)	35,220	8.70
Luxembourg	4070	49.9	50.1	13.5	1442 (30.1)	772 (16.3)	783 (19.2)	94,278	5.30
Moldova	4686	49.9	50.1	13.5	1577 (30.5)	816 (17.4)	416 (16.1)	5190	6.59
Malta	2576	51.9	48.1	13.4	819 (31.8)	795 (17.5)	2095 (44.7)	36,513	8.60
The Netherlands	4698	51.3	48.7	13.5	1480 (32.4)	876 (19.2)	517 (11.0)	48,473	10.00
Portugal	6126	52.2	47.8	13.3	1279 (33.8)	810 (19.4)	925 (15.1)	27,937	9.40
Romania	4567	51.2	48.8	13.2	1316 (34.3)	737 (19.5)	1460 (32.0)	23,313	5.60
Russia	4281	52.3	47.7	13.8	1584 (34.8)	791 (20.6)	1385 (32.4)	24,766	5.36
Ukraine	6660	51.2	48.8	13.4	1671 (35.4)	1047 (21.9)	600 (14.3)	7894	7.52
Sweden	4185	50.3	49.7	13.6	1600 (37.4)	994 (23.2)	2509 (37.7)	46,949	10.90
Slovenia	5667	49.2	50.8	13.6	10,296 (42.3)	1220 (23.6)	1418 (25.0)	31,401	8.30
Slovakia	4785	48.7	51.3	13.3	2851 (42.8)	1676 (25.2)	1649 (34.5)	30,155	6.70
UK ^b^	24,369	49.9	50.1	13.5	2049 (43.7)	6409 (26.3)	6159 (25.3)	39,753	9.70
Total	162,792	50.7	49.3		46,819 (28.8)	27,456 (16.9)	36,051 (22.1)		
Mean (SD)				13.49 (1.63)				35,552.01 (15,050.50)	8.47 (1.77)

Note. GDP per capita (PPP USD) = gross domestic product per capita (power purchasing power: USD); SD = standard deviation. ^a^ Merged French and Flemish samples (Belgium). ^b^ Merged English, Welsh, and Scottish samples (United Kingdom).

**Table 2 ijerph-21-00863-t002:** Multilevel Binary Logistic Regression Models Predicting the Association between Traditional Bullying Perpetration and Traditional Bullying Victimisation and the Moderating Role of Perceived Social Support and Health Care Spending.

	Null (Empty) Model 1	Model 2	Model 3	Model 4	Model 5
Variable	OR (95% CI)	OR (95% CI)	OR (95% CI)	OR (95% CI)	OR (95% CI)
Age (reference: 11-year-olds)		1.00 (1.00–1.00)	1.00 (1.00–1.00)	1.00 (1.00–1.00)	1.00 (1.00–1.00)
13-year-olds		0.84 *** (0.82–0.87)	0.84 *** (0.82–0.87)	0.82 *** (0.79–0.84)	0.82 *** (0.80–0.84)
15-year-olds		0.61 *** (0.59–0.63)	0.61 *** (0.59–0.63)	0.59 *** (0.57–0.61)	0.59 *** (0.57–0.61)
Gender (reference: male)		1.13 *** (1.10–1.16)	1.13 *** (1.10–1.15)	1.13 *** (1.10–1.15)	1.12 *** (1.10–1.14)
Family affluence		1.29 *** (1.25–1.32)	1.25 *** (1.22–1.29)	1.28 *** (1.24–1.31)	1.24 *** (1.20–1.27)
Traditional bullying perpetration		5.32 *** (5.18–5.46)	7.25 *** (6.87–7.64)	6.97 *** (6.70–7.25)	5.26 *** (5.05–5.48)
Family social support (FSS)			0.72 *** (0.69–0.74)		0.61 *** (0.60–0.63)
Teacher social support (TSS)				1.29 *** (1.26–1.33)	1.12 *** (1.09–1.14)
FSS * traditional bullying perpetration			0.62 *** (0.58–0.66)		
TSS * traditional bullying perpetration				0.61 *** (0.58–0.64)	
GDP per capita					0.98 * (0.97–1.00)
Health care spending (HCS)					1.05 ** (1.02–1.09)
HCS * traditional bullying perpetration					0.93 ** (0.88–0.98)
Random variances					
Country	0.10	0.05	0.06	0.05	0.06
School	0.27	0.16	0.15	0.15	0.14
Intraclass correlations (ICC)					
Country	0.02	0.01	0.01	0.01	0.01
School	0.05	0.03	0.03	0.03	0.03
Goodness-of-fit					
AIC	728,327.89	750,334.39	751,941.53	750,978.46	752,406.18
BIC	728,347.89	750,534.39	751,961.53	750,998.46	752,426.18

Note: OR = odds ratio; CI = confidence interval; AIC = Akaike information criterion; BIC = Bayesian information criterion; GDP = gross domestic product. Models were adjusted for compositional characteristics (age, gender, and family affluence) and country-level covariates (GDP per capita). *** *p* < 0.001; ** *p* < 0.01; * *p* < 0.05.

**Table 3 ijerph-21-00863-t003:** Multilevel Binary Logistic Regression Models Predicting the Association between Traditional Bullying Perpetration and Cyberbullying Victimisation and the Moderating Role of Perceived Social Support and Health Care Spending.

	Null (Empty) Model 1	Model 2	Model 3	Model 4	Model 5
Variable	OR (95% CI)	OR (95% CI)	OR (95% CI)	OR (95% CI)	OR (95% CI)
Age (reference: 11-year-olds)		1.00 (1.00–1.00)	1.00 (1.00–1.00)	1.00 (1.00–1.00)	1.00 (1.00–1.00)
13-year-olds		0.95 ** (0.92–0.99)	0.96 * (0.92–0.99)	0.95 ** (0.92–0.99)	0.95 ** (0.92–0.99)
15-year-olds		0.83 ** (0.80–0.86)	0.82 *** (0.79–0.85)	0.82 *** (0.79–0.85)	0.82 *** (0.79–0.85)
Gender (reference: male)		1.30 *** (1.26–1.34)	1.30 *** (1.27–1.34)	1.31 *** (1.27–1.34)	1.30 *** (1.26–1.34)
Family affluence		1.27 *** (1.23–1.31)	1.21 *** (1.18–1.25)	0.97 *** (0.97–0.98)	1.22 *** (1.18–1.26)
Traditional bullying perpetration		5.14 *** (5.00–5.30)	7.85 *** (7.44–8.28)	7.77 *** (7.44–8.11)	5.64 *** (5.40–5.90)
Family social support (FSS)			0.59 *** (0.56–0.61)		0.44 *** (0.42–0.45)
Teacher social support (TSS)				1.32 *** (1.28–1.38)	1.00 (0.97–1.03)
FSS * traditional bullying perpetration			0.48 *** (0.45–0.51)		
TSS * traditional bullying perpetration				0.47 *** (0.45–0.50)	
GDP per capita					0.99 (0.98–1.01)
Health care spending (HCS)					1.10 *** (1.05–1.15)
HCS * traditional bullying perpetration					0.74 *** (0.69–0.78)
Random variances					
Country	0.12	0.07	0.10	0.07	0.10
School	0.25	0.19	0.15	0.19	0.15
Intraclass correlations (ICC)					
Country	0.02	0.01	0.01	0.01	0.01
School	0.03	0.03	0.02	0.03	0.02
Goodness-of-fit					
AIC	786,937.40	816,896.16	819,944.10	817,710.64	824,447.96
BIC	786,957.40	816,716.16	819,964.10	817,730.64	824,467.96

Note: OR = odds ratio; CI = confidence interval; AIC = Akaike information criterion; BIC = Bayesian information criterion; GDP = gross domestic product. Models were adjusted for compositional characteristics (age, gender, and family affluence) and country-level covariates (GDP per capita). *** *p* < 0.001; ** *p* < 0.01; * *p* < 0.05.

## Data Availability

The current study used the 2017/18 Health Behaviour in School-aged Children (HBSC) data, which are available on request from the University of Bergen website: https://www.uib.no/en/hbscdata (accessed on 4 November 2022).

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
