# Peer review of "Perceived Social Support and Health Care Spending as Moderators in the Association of Traditional Bullying Perpetration with Traditional Bullying and Cyberbullying Victimisation among Adolescents in 27 European Countries: A Multilevel Cross-National Study"

_ijerph, 2024, doi:10.3390/ijerph21070863_

Round 1

Reviewer 1 Report

Comments and Suggestions for Authors

First of all, thank you for the opportunity to review this work.

This manuscript seeks to examine how public health expenditure and perceived social support from families and teachers moderate the relationships between the perpetration of traditional bullying and victimisation by traditional bullying and cyberbullying in 27 European countries; perceived social support from families and teachers can protect adolescents from associations between the perpetration of traditional bullying and victimisation by traditional bullying and cyberbullying.

The methodology is sound, and the results are presented clearly. This manuscript meets the journal's criteria. I will outline the main concerns in some detail in the following section.

The introduction was successfully highlighted. It cites relevant existing literature, providing a theoretical basis for the study. However, the discussion could benefit from more explicitly articulating the research gap or statement of the problem this study intends to address. A clear definition of this gap would strengthen the justification of the current research. In this regard, the authors state: 

"However, prior research has not largely focused on how proxy country-level social support (i.e., public health care spending) and each type of perceived social support (i.e., family and teacher support) moderate the association of traditional bullying perpetration with traditional bullying and cyberbullying victimisation at cross-national level since mosof the studies have used data from single countries, such as the USA, South Korea and Sweden …" (p. 4-5)  

The discussion should be explicitly more related to the existing literature, strengthening the contribution of the current study to the broader field. In addition, a more in-depth and supported exploration of the findings for educators, psychologists or others working with adolescents and, thus, the possible further implications the study seeks to assert would be useful. 

I urge the authors to implement the limitations discussed in the text. 

For example, the choice of scales and cultural and gender differences could be better discussed within the limitations.

Reviewer 2 Report

Comments and Suggestions for Authors

The authors have done an excellent work. This is an important paper. I suggest only a few changes:

-The abstract is unclear. I suggest making it shorter, clarifying the aims of this work better

- The concept and psychological impact of cyberbullying should be explored in depth in the introduction
